# Indolylazine Derivative Induces Chaperone Expression in Aged Neural Cells and Prevents the Progression of Alzheimer’s Disease

**DOI:** 10.3390/molecules27248950

**Published:** 2022-12-15

**Authors:** Vladimir F. Lazarev, Elizaveta A. Dutysheva, Elena R. Mikhaylova, Maria A. Trestsova, Irina A. Utepova, Oleg N. Chupakhin, Boris A. Margulis, Irina V. Guzhova

**Affiliations:** 1Institute of Cytology of Russian Academy of Sciences, Tikhoretsky Ave. 4, 194064 St. Petersburg, Russia; 2Department of Organic and Biomolecular Chemistry, Ural Federal University, Mira Street, 19, 620002 Ekaterinburg, Russia

**Keywords:** Alzheimer’s disease, ageing, cellular proteostasis, molecular chaperones, IA-50

## Abstract

The risk of progression of most sporadic neurodegenerative diseases, including Alzheimer’s disease, increases with age. Traditionally, this is associated with a decrease in the efficiency of cell protection systems, in particular, molecular chaperones. Thus, the development of small molecules able to induce the synthesis of chaperones is a promising therapeutic approach to prevent neural diseases associated with ageing. Here, we describe a new compound **IA-50**, belonging to the class of indolylazines and featured by a low size of topological polar surface area, the property related to substances with potentially high membrane-penetrating activity. We also estimated the absorption, distribution, metabolism and excretion characteristics of **IA-50** and found the substance to fit the effective drug criteria. The new compound was found to induce the synthesis and accumulation of Hsp70 in normal and aged neurons and in the hippocampi of young and old mice. The transgenic model of Alzheimer’s disease, based on 5xFAD mice, confirmed that the injection of **IA-50** prevented the formation of β-amyloid aggregates, loss of hippocampal neurons and the development of memory impairment. These data indicate that this novel substance may induce the expression of chaperones in neural cells and brain tissues, suggesting its possible application in the therapy of ageing-associated disorders.

## 1. Introduction

The progression of many neurodegenerative diseases is explained by the formation of toxic oligomers or aggregates consisting of mutant or damaged proteins that are located inside or in close proximity to neurons. The accumulation of such pathogenic structures disturbs the normal functioning of metabolism and signalling and leads to dysfunction and, ultimately, death of neural cells [1]. According to the amyloid hypothesis of Alzheimer’s disease (AD), cytotoxic oligomers and aggregates of Aβ peptide contribute to the progression of the disease [2,3]. Of note, AD as well as many other neurological diseases, progression with age may be due to the reduced ability of the cell protection systems to resist multiple damages in cellular proteostasis. Molecular chaperone Hsp70 constitutes one of the aforementioned protection systems since it has been shown to bind mutant, or damaged proteins/peptides, and prevent their oligomerisation and aggregation [4,5]. In a view of AD pathogenesis, the chaperones were found to target mutant tau and β-amyloid and prevent the formation of their cytotoxic aggregates in AD pathogenesis [6,7].

One of the ways to activate the previously discussed mechanisms is the application of chemical inducers of chaperone synthesis [8], whose protective activity has been proven in a few neurodegeneration models, including AD [9]. However, the bottleneck of this therapeutic approach is that the efficiency of chaperone induction and its cellular concentration is reduced with ageing [10,11]. However, suppression of Hsp70 content and other systems of cellular proteostasis with age has been shown for rat hepatocytes [12], rat and rhesus monkey lymphocytes [13] and human lymphocytes [14,15]. Although it is a common thread in many reviews that the proteostasis system deteriorates with age, we did not find much experimental information about chaperone expression in brain structures, and the data presented in these studies contradict those in other tissues. In the first study, it was demonstrated that both Hsp70 and Hsc70 (constitutive members of the same protein family) increased in the hippocampus, substantia nigra, cerebellum, cortex and striatum during ageing due to the significant changes in the glutathione redox state and lipid peroxidation [16]. In another paper, the authors did not find any difference in Hsp60, Hsp40 or Hsc70 expression in the substantia nigra and striatum during the life span of female rats [17].

The aim of this work was (i) to evaluate the proteostasis status in aged brain structures involved in AD pathogenesis and (ii) to search for the compounds that cause the accumulation of chaperone Hsp70 in older cells or tissues. 

Previously, we found that substances with a certain general structure (belonging to pyrrolyl- and indolylazines groups of chemicals) were able to induce the synthesis and accumulation of chaperones, which led to a therapeutic effect in an in vitro model of AD [18]. In the present study, based on in silico calculations, we selected, synthesised and tested a new compound from the indolylazine series, **IA-50**, which activated the synthesis of the Hsp70 in a culture of aged human neural cells. This compound was also tested on an in vivo model of 5xFAD transgenic mice, which were producing β-amyloid and demonstrating general manifestations of AD.

## 2. Results

### 2.1. Analysis of Proteins Regulating Cell Proteostasis in Ageing 

The inconsistent data on the regulation of the proteostasis system in ageing brains forced us to carry out an analysis of proteins involved the protein folding control and degradation. We selected mice at the age of 3, 6, 9, 12, 18 and 24 months (*n* = 5 in each group). To understand what is happening in those regions of the brain where the primary focus of neurodegeneration is located, namely in the hippocampus, as well as in the cerebral cortex, suffering in more advanced stages of the disease, we performed Western blotting of the hippocampal and cortical fractions, using antibodies for the selected proteostasis actors. 

The results of Western blotting convinced us that the efficiency of proteostasis systems decreases over the course of a lifetime (Figure 1). The level of molecular chaperones Hsp70 and Hsp40, two proteins functioning in tandem that define the fate of cellular proteins with abnormal conformation, both in the hippocampus and in the cortex, began to decrease after 12 months, and we observed the same pattern for Hsc70, a protein that plays both the role of a cellular chaperone and the main participant in Chaperone mediated autophagy (CMA). Furthermore, in both analysed brain structures, starting from 9–12 months of age, a reduction in the expression of the autophagy marker, Atg5/12, was observed (Figure 1). These data prompted us to search the compounds able to force at least molecular chaperone expression in the aged brains of AD mice.

### 2.2. In Silico Prediction of the Indolylazine Ability to Penetrate the Cell Membrane of Senescent Neurons 

Previously, we established the chemical structure necessary to activate the synthesis and accumulation of molecular chaperones in neural cells. However, most of the chemicals were unable to efficiently induce chaperone synthesis in senescent cells or aged mice (Lazarev et al., unpublished data). This insufficiency was presumably related to changes in membrane permeability in aged cells, due to reduced amounts of polyunsaturated fatty acids [19,20]. One of the key parameters, which is considered to reflect the ability of small molecules to penetrate the cell membrane, is the topological polar surface area (TPSA). The lower the TPSA value—the higher probability of passing a small molecule through the cell membrane [21]. Due to the fact that the compounds synthesized and tested by us did not have a significant effect on the chaperone synthesis in senescent cells, we conducted an in silico analysis of the physicochemical properties of a few of the indole derivatives synthesized by us and compared them with those previously published [18]. We noticed that one of the new compounds, **IA-50**, stands out with the lowest TPSA value (Table 1), which gave hope for high membrane-penetrating activity of this substance. Of significance, comparing the structure of **IA-50** with other relative molecules (Figure 2), we found that it retains the features of heat response—inducing activity and selected the compound for further examinations of its biological properties.

### 2.3. **IA-50** Is Able to Induce Chaperone Accumulation in Aged Neurons

Since the major goal of this report was to develop a heat shock response inducer in aged neurons, we used human umbilical cord mesenchymal stem cells MSCWJ-2 and reprogrammed them to the neuronal phenotype after the 5th and 15th passages. At the 15th passage, the MSCWJ-2 cells demonstrated senescent phenotype as was proved by approximately 2-fold higher β-galactosidase activity compared to cells of the fifth passage (Figure 3A). Notably, both cells at the 5th and 15th passages were reprogrammed into neurons, that was confirmed by vastly elevated expression of β-3 tubulin and MAP-2 (Figure 3B).

Using this model, we tested the ability of **IA-50** to induce gene expression of the Hsp70 chaperone in MSCWJ-Neu cells. This was performed with the aid of RT-PCR, using mRNA isolated from young MSCWJ-Neu (5 passages) and aged MSCWJ-Neu (15 passages) cells cultured in normal conditions and after a 6-h treatment with **IA-50** at a concentration of 1 µM. We found that the amount of mRNA of the Hsp70 protein gene increased 2-fold in young cells and 1.2-fold in senescent cells (Figure 3C).

Then, we used Western blotting to measure the Hsp70 concentration in young and aged MSCWJ-Neu cells and found that **IA-50** was able to cause Hsp70 accumulation in both types of cells (Figure 3D). To estimate the Hsp70-inducing effect in brain tissue, we injected **IA-50** (10 µg/kg) in 3- and 16-month-old mice, and 72 h later, the animals were sacrificed, the hippocampi were isolated and subjected to Western blotting. As revealed by Western blotting, **IA-50** was able to cause Hsp70 accumulation in the hippocampi of both young and aged mice, suggesting that the substance is able to increase Hsp70 in vitro and in vivo to a similar extent in older and younger animals (Figure 3E).

### 2.4. The Treatment of 5xFAD Mice with **IA-50** Improves Memory Defects and Aβ42 Aggregation

The hallmarks of major AD causalities are paralleled with brain ageing, and therefore, we tested the therapeutic activity of **IA-50** using 5xFAD mice, a transgenic model of AD. At the P180 time point, when, according to the literature data, signs of AD can already be observed, such as amyloid deposits in the hippocampus [22,23], the animals were divided into four groups: (1) ‘untreated wt’ mice (*n* = 10); (2) wt mice treated with **IA-50** (‘wt-IA-50’) (*n* = 10); (3) ‘untreated 5xFAD’ mice (*n* = 10); (4) 5xFAD mice treated with **IA-50** (‘5xFAD-IA-50’) (*n* = 10). The drug was administered intramuscularly at a dose of 10 mg/kg once a week, and at P360, mice were subjected to different behavioural tests.

Anxiety is considered to be a sign of early AD which progresses throughout the AD pathogenesis due to brain damage and psychosocial factors [24,25]. The marble burying test was employed for the screening of new antidepressants, anxiolytics and antipsychotics [26,27], and the assay was recently found to reveal that 3xTg-AD and Tg-APP/PS1 mice were much less efficient in burying marbles than their healthy peers [28,29]; this prompted us to apply this test in our study. In these experiments, the marbles were placed on the sawdust of the cage bottom as depicted in Figure 4A and left each mouse with marbles for 30 min. Healthy wt mice successfully buried marbles irrespective of whether or not they were treated with **IA-50**; eight mice in the ‘untreated wt’ group were able to hide all the marbles and two buried only ¾ marbles. ‘Wt-IA-50’ demonstrated similar capacities: seven mice buried all the marbles and three only buried ¾. In the ‘untreated 5xFAD’ group, only three mice were able to hide ¾ of the marbles but treatment with **IA-50** increased the mice’s capacity to bury marbles; two mice from this group buried all the marbles, and five hid ¾ of the marbles (Figure 4B).

Another well-established behavioural assay, the Morris water maze test, was performed at time point P450 (15 months of age) when we expected a more pronounced effect of ageing and of Hsp70-inducing compounds. The 5xFAD mice that did not receive the **IA-50** treatment searching time for the platform was 2.5-fold longer than that of their healthy siblings. The 5xFAD mice treated with **IA-50** were able to find the platform 2-fold faster than untreated 5xFAD mice (Figure 4C). The tracks that the 5xFAD mice chose in search of the platform were significantly shorter than those of untreated 5xFAD mice, although their tracks were longer than those in their healthy siblings, independently as to whether or not they received the **IA-50** treatment (Figure 4D).

The animals that passed the Morris maze test were sacrificed and their brains were divided into two parts, one was subjected to the histochemical study, and another was used for biochemical tests. We analyzed the frontal sections at the level of the CA1 field of the hippocampus and found that the number of neurons (according to staining for the NeuN marker) in the 5xFAD mice was 57% less than in the healthy ones. At the same time, the loss of neurons in the 5xFAD mice treated with **IA-50** was significantly less—only 22% (Figure 5A,B). Finally, using the method of ultrafiltration followed by hybridization with anti-Aβ antibodies, we estimated the accumulation of Aβ42 aggregates in the hippocampus of animals related to different groups. We found a significant number of Aβ42 aggregates in the hippocampus of the 5xFAD mice, while there were no aggregates in the healthy mice. The use of **IA-50** resulted in a decrease in the number of aggregates in transgenic animals by approximately 30% (Figure 5C,D).

## 3. Discussion

It is well established that the vast majority of human pathologies transform into more dangerous forms with age, partially due to an inability of numerous protein homeostasis systems to cope with the increasing number of damaged or mutant proteins [30]. The reduction of that capacity was suggested to be particularly negative for molecular chaperones and autophagy; chaperones are able to recognize pathogenic polypeptides through exposed hydrophobic surfaces and convert them into harmless forms or to assist proteolytic machinery to finally degrade the substrate [31]. However, the data on Hsp70 expression in young and aged cells and tissues are very contradictory: there are proofs of enhanced expression of Hsps in aged cells, probably due to their permanent exposure to stress [9,16] or reduced level of chaperones due to damages in transcription/translation machinery [32]. Our analysis of cell proteostasis actors in mice of various ages starting with 3-month-old mice and terminating with 24-month-old mice demonstrated that levels of Hsp70 chaperone and its co-chaperone Hsp40 (Hdj1 or DNAJB1), as well as Hsc70 which could function as a chaperone [33] and a main actor of CMA [34], and autophagy marker Atg5/12, significantly decrease at approximately 12 months of age and decline further over the next 12 months.

Previously, we analysed a few pyrrolyl- and indolylazine compounds and proved their ability to induce a heat shock response [18]. Here, we performed a bioinformatic search aimed at finding the substance with the maximum ability to penetrate living cells with a possibly modified lipid membrane due to ageing [35]. Such a compound, indolylazine **IA-50**, was able to induce Hsp70 chaperone synthesis in both young and aged reprogrammed human neurons MSCWJ-NEU.

Furthermore, administration of **IA-50** was also found to induce Hsp70 expression in the hippocampi of mice, suggesting that the effect remained minimal one day after the **IA-50** injection.

It should be noted that the development of new approaches to the synthesis of bio- and pharmaceutically active compounds has become widespread among organic chemists in order to improve traditional synthetic methods. In this regard, photocatalytic transformations are a topic of interest, for both scientists and practitioners, since it is an environmentally friendly and available alternative to chemical oxidizing and reducing agents [36,37]. An **IA-50** derivative was prepared by the C–H/C–H coupling of acridine with 4,5,6,7-tetrahydro-1*H*-indole in acetic acid at room temperature, using an effective heterogeneous air oxygen/TiO_2_ oxidative system. All reactions were carried out in a quartz flask under irradiation with a Xe lamp (5000 K, 35 W) for 5 h, according to our previously developed methodology [38].

The treatment with **IA-50** was started when the 5xFAD mice were six months old because it was shown to be possible to detect morphological symptoms of AD [39]. The therapy was carried out for the next 9 months, once a week, until the mice were 12 months old; according to Rae et al., this age corresponds to the upper limit of the average age (50–60 years) in humans [39]. The marble burying assay could testify to the elevated anxiety, which is considered to be a sign of early AD [24,25]. The 5xFAD mice were not able to bury marbles; whereas, their healthy siblings demonstrated successful burying efficacy independently of whether or not they were treated with **IA-50**. However, the results from the 5xFAD mice treated with **IA-50** significantly improved in this test. Three months later, when the mice’s ages corresponded to old age in humans (~70–90 years in humans), mice passed through the Morris water maze test; the results of which predictably showed the preservation of spatial memory in the 5xFAD mice treated with **IA-50**. The data from the immunochemistry and staining with anti-NeuN and anti-amyloid antibodies demonstrated that the **IA-50** treatment provided the preservation of the hippocampal neuron number in 5xFAD mice and a reduced number of b-amyloid aggregates. The ultrafiltration assay confirmed the results of the immunochemistry experiments.

To date, the list of Hsp70 inducers includes celastrol, arimoclomol [40], geranylgeranyl-acetone [41], and U-133 [42]; the compounds have undergone pre-clinical trials, and as far as we know, the latter has been subjected to clinical trials as the medicine for orphan lysosome storage disease [43]. We found that **IA-50** is capable of coping with the most established attributes of AD pathogenesis and, therefore, believe that the compound, after improving its properties, including solubility and TPSA, may be considered as a candidate for pre-clinical trials.

Similar studies on pre-aged neurons have not been previously performed; nevertheless, it is noteworthy that, according to our data, the response of aged cells to the **IA-50** compound was indeed less pronounced compared to the response of young cells. Certain limitations in the use of pyrrolyl- and indolylazine derivatives as neuroprotectors are currently seen in their low water solubility. For this reason, it is not possible to increase the working concentration of the compound, which, given the extremely low toxicity, could be useful. Probably, when conducting further searches for leader compounds from the classes of pyrrolyl- and indolylazines, it will be necessary to focus on compounds with low TPSA values (<40 A) and high water solubility values (Log S > −2).

## 4. Materials and Methods

### 4.1. In Silico Assays

The preliminary in silico assessment of pharmacokinetic properties for absorption, distribution, metabolism and excretion properties (ADME) was performed with the SwissADME tool developed by the Swiss Institute of Bioinformatics [44]. The computational analysis included predictions of molecular descriptors, such as solubility, topological polar surface area, gastrointestinal absorption and blood–brain barrier permeability. The SwissADME toolkit contains classification models, which focus on the tendency for a given small molecule to serve as an inhibitor of proteins governing important pharmacokinetic behaviours. Drug-likeness was assessed under Lipinski’s rule of five, which is based on descriptors such as molecular weight, number of hydrogen bond acceptors, number of hydrogen bond donors and octanol/water partition coefficient.

### 4.2. Synthesis of 9-(1H-indol-2-yl)acridine (**IA-50**)

The starting 4,5,6,7-tetrahydroindole is a compound available on an industrial scale and can be easily obtained by the reaction of cyclohexanone oxime with acetylene in the superbasic catalytic system KOH/DMSO [45].

A quartz flask containing a solution of acridine 1 (1 mmol), 4,5,6,7-tetrahydro-1*H*-indole 2 (2 mmol) in acetic acid (20 mL) and a TiO_2_ catalyst (10 wt.%) was sonicated for 5 min to obtain a suspension. The resulting mixture was irradiated with a Xe lamp (5000 K, 35 W) under air bubbling at room temperature (20 °C). The reaction was completed after 5 h. The reaction mass was concentrated under reduced pressure. The resulting residue was subjected to preparative SiO_2_ column chromatography. A mixture of hexane:ethyl acetate (8:2) was used as the eluent. The resulting compound of 3 (9-(4,5,6,7-tetrahydro-1*H*-indol-2-yl)acridine) and DDQ was dissolved in toluene (30 mL) and boiled for 8 h. The solution was concentrated under reduced pressure. The resulting residue was subjected to preparative SiO_2_ column chromatography (benzene) (Figure 1).

9-(4,5,6,7-tetrahydro-1*H*-indol-2-yl)acridine (**3**). Yield: 149 mg (50%); yellow solid; *R_f_ =* 0.1 (TLC: hexane:ethyl acetate, 8:2). ^1^H NMR (400 MHz, DMSO-*d*_6_): *δ* = 11.12 (s, N’-H), 8.21–8.23 (d, *J* = 8.00 Hz, 1-H, 8-H), 8.14–8.16 (d, *J* = 8.00 Hz, 4-H, 5-H), 7.81–7.84 (m, 3-H, 6-H), 7.55–7.59 (t, *J* = 8.00 Hz, 2-H, 7-H), 6.26–6.27 (m, 3′-H), 2.69–2.72 (m, 4′-H), 2.61–2.64 (m, 7′-H), 1.81–1.86 (m, 5′-H, 6′-H). ^13^C NMR (100 MHz, DMSO-*d*_6_): *δ* = 148.39 (C4a, C9a), 139.56 (C9), 129.97 (C3, C6), 129.06 (C1, C8), 127.36 (C4, C5), 125.54 (C2, C7), 124.66 (C8a, C10a), 121.84 (C2′), 117.31 (C3a’, C7a’), 112.03 (C3′), 23.50 (C4), 23.04 (C5), 22.56 (C6), 22.05 (C7). MS (EI): m/z 298 [M]^+^. Anal. Calcd for C_21_H_18_N_2_ (%): C, 84.82; H, 5.76; N, 9.42. Found (%): C, 84.76; H, 5.83; N, 9.71.

9-(1H-indol-2-yl)acridine (**IA-50**). Yield: 102 mg (35%); yellow solid; *R_f_ =* 0.1 (TLC: benzene); mp 292 °C. ^1^H NMR (400 MHz, DMSO-*d_6_*): δ = 11.84 (1 H, NH), 8.25 (d, *J* = 8.6 Hz, 2H, 3-H, 6-H), 7.99 (d, *J* = 8.6 Hz, 2H, 1-H, 8-H), 7.89 (t, *J* = 7.3 Hz, 2H, 2-H, 7-H), 7.74 (d, *J* = 7.7 Hz, 1H, 4′-H), 7.60 (t, *J* = 7.3 Hz, 2H, 4-H, 5-H), 7.52 (d, *J* = 7.8 Hz, 1H, 6′-H), 7.24 (d, *J* = 7.8 Hz, 1H, 5′-H), 7.16 (t, *J* = 7.0 Hz, 1H, 7′-H), 6.88 (s, 1H, 3′-H). ^13^C NMR (100 MHz, DMSO-*d_6_*): δ = 148.75 (C8a, C9a), 138.89 (C9), 137.48 (C3a’), 131.37 (C2′), 130.80 (C2, C7), 129.82 (C3, C6), 128.51 (C7a’), 127.14 (C1, C8), 126.95 (C4, C5), 125.61 (C4a, C10a), 122.48 (C5′), 120.82 (C4′), 120.12 (C7′), 111.99 (C6′), 106.02 (C3′).

NMR, NSQC and HMBC spectrums of **IA-50** are presented in Appendix A.

### 4.3. Mimicking the Neuronal Cell Phenotype

For further verification of the chaperone-inducing and neuroprotective properties of PLAs, we used mesenchymal stem cells from Wharton’s jelly of human umbilical cord (MSCWJ-2) cells that was described earlier [46]. The cells were received from the shared research facility “Vertebrate cell culture collection” supported by the Ministry of Science and Higher Education of the Russian Federation (Agreement NO. 075-15-2021-683). Cells were cultured in DMEM/F12 medium (Gibco, Paisley, UK), containing 10% fetal bovine serum (FBS; Gibco, Paisley, UK), 100 units/mL penicillin and 0.1 mg/mL streptomycin (BioloT, St.Petersburg, Russia) at 37 °C and 5% CO_2_. We reprogrammed these cells into a neuronal phenotype for 5 days in a Neurobasal medium (Gibco, Paisley, UK), containing B27 supplement (Gibco, Paisley, UK), 3% FBS, 100 units/mL penicillin and 0.1 mg/mL streptomycin. Verification of the neuronal phenotype was carried out based on the analysis of the expression of neuronal markers MAP-2 and β-3-tubulin, using a real-time polymerase chain reaction. The cells were named MSCWJ-NEU.

### 4.4. RNA Isolation and Real-Time PCR

RNA was isolated using TRIzol (Thermo Fisher Scientific, Waltham, MA, USA) and reverse transcribed, using the MMLV RT kit (Evrogen JSC, Moscow, Russia) according to the manufacturer’s instructions. All RT-PCR reactions were performed with a CFX96 Real-Time PCR detection system (BioRad, Hercules, CA, USA), using qPCRmix-HS SYBR (Evrogen JSC, Moscow, Russia) and according to the manufacturer’s protocol. Amplicon authenticity was confirmed by melt curve analysis. The sequence of primers is represented in Table 2.

GAPDH was used as the normalisation control. All primers were obtained from Evrogen JSC (Moscow, Russia). The parameters of the polymerase chain reaction (PCR) were 5 min of pre-denaturation at 95 °C, followed by 40 cycles of 30 s at 95 °C, 30 s at 65 °C and 30 s at 70 °C. The data were analysed for fold changes (ΔΔCt) using BioRad CFX software.

### 4.5. Electrophoresis and Immunoblotting

MSCWJ-NEU cells were incubated with **IA-50** at a concentration of 1 µM for 24 h. Cells were then lysed by three freeze–thaw cycles in a low RIPA buffer containing 20 mM Tris-HCl pH 7.5, 150 mM NaCl, 0.05% Tween-20, 0.1% SDS, 3 mM EDTA and 1 mM PMSF. To obtain lysates of hippocampal tissues, the mice were sacrificed, and the hippocampus was immediately removed and lysed in a low RIPA buffer. Lysates of MSCWJ-NEU cells or hippocampal tissues were subjected to electrophoresis, following Western. The membrane was subsequently incubated with antibodies against Hsp70, clone 3B5 [47] and glyceraldehyde-3-phosphate dehydrogenase, taken as a loading control (GAPDH, Clone 6C5, Abcam, Cambridge, UK).

### 4.6. Senescence Analysis

To measure β-galactosidase activity in cells, X-Gal staining solution (Thermo Fisher Scientific, Waltham, MA, USA) was added to the cell culture, according to the manufacturer’s protocol. After 72 h, microscopic photos were collected and analyzed, using ImageJ software, in order to measure the colorimetric staining intensity.

### 4.7. Transgenic Mice Model of AD

The 5xFAD transgenic mice were genotyped by PCR analysis of DNA extracted from the ear biopsies. The transgenic cassette was detected, using primers 5′-AGGACTGACCACTCGACCAG-3′ and 5′-CGGGGGTCTAGTTCTGCAT-3′, yielding a 377 bp product. Siblings of 5xFAD mice were used as a wild-type (WT); this protocol was described by Peters et al. [48].

WT and 5xFAD male mice at P180 were divided into 4 groups of 10 animals each as follows: (1) WT, untreated; (2) WT, treated with **IA-50**; (3) 5xFAD, untreated; (4) 5xFAD, treated with **IA-50**. Treatment involved intraperitoneal injections of **IA-50** once a week at a dosage of 1 mg/kg within an additional 180 days.

### 4.8. Animal Behaviour Tests

All in vivo experiments were carried out following the requirements of the Institute of Cytology of the Russian Academy of Sciences ethics committee (Identification number F18-00380).

5xFAD mice were purchased in the Center of Animal Models in the Institute of Physiologically Active Compounds of Russian Academy of Sciences (Chernogolovka, Moscow Region, Russia). The model is maintained by backcrossing transgenic animals to a B6SJLF1 hybrid at every generation.

#### 4.8.1. Marble Burying Assay

The procedures were performed as described earlier [49] with modification. Briefly, a plastic cage (40 cm × 20 cm × 17 cm) was filled with sawdust up to 5 cm deep, and 12 marbles were placed on top of the sawdust, 3 marbles in four rows (see Figure 3A). Each mouse was placed in the cage and was left with the marbles for 30 min; then, the mouse was removed and the number of buried marbles was counted.

#### 4.8.2. Morris Water Test

Spatial memory impairment in 5xFAD mice from each experimental group (*n* = 10) was evaluated, using a Morris water maze test [50] of diameter 1.5 m (OpenScience, Krasnogorsk, Russia) on P450. Mice were trained to find the platform for 5 days. Daily training included 4 attempts with an interval of 30 s. During these attempts, the animal was sequentially placed in different sectors (the sequence of sectors was determined randomly). If the mouse did not find the platform, it was forcibly placed on it. In any case, the animal was left on the platform for 15 s. Animal movements were recorded for a minute with a digital video camera, which was located at a height of 450 cm from the floor and connected to a computer via a USB interface. On the 6th day, spatial memory was tested. The mice had one attempt to find the platform. The animal was placed in the pool at the farthest sector from the target sector.

The MWMtrack-9 software was used to build motion tracks and evaluate memory. When evaluating the spatial memory of the mice, the latent release time (s), during which the mouse found the platform and climbed onto it, was calculated. If the animal did not find the platform, then the value of the latent time was taken to be equal to 60 s. The effectiveness of training was assessed by reducing the time spent on the platform and the reduction of movement tracks. The swimming trajectories of animals in experiments were treated according to EthoVision XT14.0 [51].

### 4.9. Immunohistochemistry

At the end of the Morris water test, the mice were anaesthetized with Zoletil-100 (50 mg/kg, intraperitoneal), perfused with 4% paraformaldehyde and then decapitated. The brain was extracted and examined by confocal microscopy. Brains from all animals used for immunohistochemical assays were fixed in 4% paraformaldehyde and cryoprotected in 20% sucrose before storage in isopentane at −70 °C. Coronal sections (8 μm) were prepared for morphological and immunohistochemical assay with an OTF 6000 cryostat (Bright Instruments, Huntingdon, UK). The frontal slices were collected at the level of the field CA1 of the hippocampus, according to The Mouse Brain Library service (mbl.org, Tennessee, USA). Six alternate series of sections were mounted on SuperFrost Plus slides (Menzel GmbH, Bielefeld, Germany).

For confocal microscopy, sections were pre-incubated in blocking solution (2% bovine serum albumin diluted in PBS with 0.1% Tween-20) for 1 h at room temperature and then hybridized with anti-NeuN (Abcam, Cambridge, UK) or anti-Aβ1-42 antibodies (Elabscience, Houston, TX, USA), following hybridization with secondary fluorescently labelled antibodies (ThermoFisher Scientific, Waltham, MA, USA). Fluorescent images were captured by an Olympus confocal system FV3000 (Olympus, Tokyo, Japan).

### 4.10. Aggregation Assay

To analyse the amount of aggregates containing Aβ42 in the mouse hippocampus, we used the ultrafiltration method (filter trap assay) previously described [52]. Mouse hippocampal lysates were dissolved in buffer (10 mM Tris-HCl pH 8.0, 150 mM NaCl, 2% sodium dodecyl sulfate) in the amount of 200 µg of total protein, applied to a cellulose acetate membrane, and placed in an ultrafiltration manifold attached to a vacuum pump (BIO-RAD, Hercules, CA, USA). Before and after applying the lysates, the membrane was washed under pressure with a buffer of the following composition: 10 mM Tris-HCl pH 8.0, 150 mM NaCl and 0.1% sodium dodecyl sulfate. The presence of Aβ1-42 in the aggregates was determined using specific antibodies, MAA946Ge21 (Elabscience, Houston, TX, USA), followed by hybridisation with secondary antibodies and labelled with peroxidase (1:10,000; Jackson Laboratory, Bar Harbor, ME, USA). Using TotalLabQuant 1.0 software (TotalLab, Gosforth, UK), we obtained the dot intensity value in conventional units, and then normalized the data to the mean Aβ42 staining intensity in the hippocampus of naive rats.

### 4.11. Statistical Analysis

All data were expressed as mean ± standard deviation. Data were compared using a non-parametric Mann–Whitney test, using GraphPad Prism 8 software. All experiments, except for animal studies, were repeated at least three times. The statistical difference was determined by *p* < 0.05.

## 5. Conclusions

In this work, we synthesised and tested a new compound from the class of indolylazines, **IA-50**, capable of activating the synthesis and inducing the accumulation of Hsp70 chaperone, both in the culture of aged human neurons and in the hippocampus of old mice, for the first time. The new compound demonstrated therapeutic potential in a transgenic model of AD in the 5xFAD mice.

## Data Availability

Not applicable.

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
