# Peer review of "Indolylazine Derivative Induces Chaperone Expression in Aged Neural Cells and Prevents the Progression of Alzheimer’s Disease"

_molecules, 2022, doi:10.3390/molecules27248950_

Round 1
Reviewer 1 Report
Paragraph 2.2 should be deeply revised and rewritten cause it's not clear and it does not ensure the reader why compound IA-50 was chosen for all the studies. The authors said that they estimated TPSA for newly synthesized indolylazine derivatives and that they choose compound IA-50 with the lowest TPSA for their study. But is not said how many compounds were analysed. In Fig. 2 we see 5 compounds, whereas in Table 1. there are 7.
Table 1 should e carefully edited! In some places there are commas and in other dots, in some places logS value has two digits after the decimal point and in other one. Under the Table there is abbrev. 'NRB' which is not used in the Table, while in Table 1. is abbrev. 'GI ' that is not under the Table.
Minor English revision should be done cause there are sentences that are incorrect, eg. Line 32 or 75.
Author Response
First, we would like to thank the reviewer for the careful reading our manuscript and for the comments that helped us to improve the m/s.
Reviewer 1.
Q1: Paragraph 2.2 should be deeply revised and rewritten cause it's not clear and it does not ensure the reader why compound IA-50 was chosen for all the studies. The authors said that they estimated TPSA for newly synthesized indolylazine derivatives and that they choose compound IA-50 with the lowest TPSA for their study. But is not said how many compounds were analysed. In Fig. 2 we see 5 compounds, whereas in Table 1. there are 7.
R1: We carefully rewrote paragraph 2.2 according to the reviewer recommendations in order to prove the benefits of IA-50 as a potential inducer of Hsp70 synthesis in aging cells. We present here data for five pyrrolyl- and indolylazines that we ever tested as Hsp70 inducers.
Q2: Table 1 should e carefully edited! In some places there are commas and in other dots, in some places logS value has two digits after the decimal point and in other one. Under the Table there is abbrev. 'NRB' which is not used in the Table, while in Table 1. is abbrev. 'GI ' that is not under the Table.
R2: We apologize for the negligence in the design of table 1. We corrected the table, left only 5 substances and aligned the designations in the table and in the list of abbreviations
Q3: Minor English revision should be done cause there are sentences that are incorrect, eg. Line 32 or 75
R3: Thank you! We have corrected
Reviewer 2 Report
The current manuscript has identified a indolylazine derivative IA-50 inducing the chaperone expression and preventing the AD phenotype. They have used appropriate methods and produced results to bolster IA-50 as a potential compound for treating AD phenotype. However, the manuscript needs to go through major revision before accepting for publication.
1. The manuscript needs to go through extensive english editing. For example: line 96 figure 1 title: Expression of protein homeostasis (proteostasis) actors in the hippocampi and cortexes during a 96 mouse’s life span.
2. In figure-5A authors need to mention the antibodies used.
3. Authors need to mention the age of the mice 5xFAD mice they used for the behavioral testing.
4. In figure1 Authors have shown age dependent reduction in the protein homeostasis factors. Is it evident that IA-50 has restored these proteins in aged 5xFAD mice?
5.In figure 4 A in Marble burying assay figure for WT mice treated IA-50 is missing.
6. Authors did not mention source of mice procurement in their methodology.
Author Response
First, we would like to thank the reviewer for the careful reading our manuscript and for the comments that helped us to improve the m/s.
Q1: The manuscript needs to go through extensive english editing. For example: line 96 figure 1 title: Expression of protein homeostasis (proteostasis) actors in the hippocampi and cortexes during a 96 mouse’s life span.
R1: To tell the truth, before sending the manuscript for evaluation in Molecules, we sent it for proof-reading at an authorized service in the UK. Apparently, some little flaws always remain. We ask you to forgive us for them. And we have changed the title of Fig.1.
Q2: In figure-5A authors need to mention the antibodies used.
R2: We apologize for the negligence in the design of figure 5. We inserted the names of the proteins (or antibodies used) shown in fig. 5A
Q3: Authors need to mention the age of the mice 5xFAD mice they used for the behavioral testing.
R3: Treatment with IA-50 started when 5xFAD mice were 6 months age (at P180) when signs of AD such as amyloid deposits in the hippocampus, were already observed; the Marble test was performed when the mice were 12 months old (at P360). The Morris water maze was performed when the mice were almost 15 months old (at P450). This information is presented in the manuscript in the Results (line 234, 239, 274) and Figure 4 legend
Q4: In figure1 Authors have shown age dependent reduction in the protein homeostasis factors. Is it evident that IA-50 has restored these proteins in aged 5xFAD mice?
R4: Since our study focused mainly on Hsp70, we tested only the possibility of increased accumulation of Hsp70 in the hippocampus of mice treated with f IA-50 and showed that the quantity of the chaperone elevated in aged mice compared with younger ones. For didactic reasons, we have added a new graph to Figure 3 showing ratio Hsp70/GAPDH (Fig3F).
Q5: In figure 4 A in Marble burying assay figure for WT mice treated IA-50 is missing.
R5: We have added the panel with wt mice treated with IA-50 to Fig. 4A.
Q6: Authors did not mention source of mice procurement in their methodology.
R6: We have added this information to Materials and methods 2.7
Round 2
Reviewer 2 Report
I do not have any other queries, as authors mentioned typos check may be necessary. manuscript can be accepted.